# CAPTURING HUMAN CATEGORY REPRESENTATIONS BY SAMPLING IN DEEP FEATURE SPACES

## ABSTRACT

Understanding how people represent categories is a core problem in cognitive science, with the flexibility of human learning remaining a gold standard to which modern artificial intelligence and machine learning aspire. Decades of psychological research have yielded a variety of formal theories of categories, yet validating these theories with naturalistic stimuli remains a challenge. The problem is that human category representations cannot be directly observed and running informative experiments with naturalistic stimuli such as images requires having a workable representation of these stimuli. Deep neural networks have recently been successful in a range of computer vision tasks and provide a way to represent the features of images. In this paper, we introduce a method for estimating the structure of human categories that draws on ideas from both cognitive science and machine learning, blending human-based algorithms with state-of-the-art deep representation learners. We provide qualitative and quantitative results as a proof of concept for the feasibility of the method. Samples drawn from human distributions rival the quality of current state-of-the-art generative models and outperform alternative methods for estimating the structure of human categories.

## 1 INTRODUCTION

Categorization (or classification) is a central problem in cognitive science (Cohen & Lefebvre, 2005), artificial intelligence, and machine learning (Duda et al., 1973). In its most general form, the categorization problem concerns why and how we divide the world into discrete units (and various levels of abstraction), and what we do with this information. The biggest challenge for studying human categorization is that the content of mental category representations cannot be directly observed, which has led to development of laboratory methods for estimating this content from human behavior. Because these methods rely on small artificial stimulus sets with handcrafted or low-dimensional feature sets, they are ill-suited to the study of categorization as an intelligent process, which is principally motivated by people's robust categorization performance in complex ecological settings.

One of the challenges of applying psychological methods to realistic stimuli such as natural images is finding a way to represent them. Recent work in machine learning has shown that deep learning models, such as convolutional neural networks, perform well on a range of computer vision tasks (LeCun et al., 2015). The features discovered by these models provide a way to represent complex images compactly. It may be possible to express human category structure using these features, an idea supported by recent work in cognitive science (Lake et al., 2015; Peterson et al., 2016).

Ideally, experimental methods could be combined with state-of-the-art deep learning models to estimate the structure of human categories with as few assumptions as possible and while avoiding the problem of dataset bias. In what follows, we propose a method that uses a human in the loop to directly estimate arbitrary distributions over complex feature spaces, adapting a framework that can exploit advances in deep architectures and computing power to increasingly capture and sharpen the precise structure of human category representations. Such knowledge is crucial to forming an ecological theory of intelligent categorization behavior and to providing a ground-truth benchmark to guide and inspire future work in machine learning.

## 2 BACKGROUND

### 2.1 ESTIMATING THE STRUCTURE OF HUMAN CATEGORIES

Methods for estimating human category templates have existed for some time. In psychophysics, the most popular and well-understood method is known as *classification images* (Ahumada, 1996). In this experimental procedure, a human participant is shown stimuli from two classes, A and B, each with white noise overlaid, and asked to indicate the correct label. On most trials, the participant will select the exemplar generated from the category in question. However, if the added white noise significantly perturbs features of the image important to making that distinction, they may fail. Exploiting this, we can estimate the decision boundary from a number of these trials using the simple formula:

$$(n_{AA} + n_{BA}) - (n_{AB} + n_{BB}), \tag{1}$$

where $n_{XY}$ is the average of the noise across trials where the correct class is $X$ and the observer chooses $Y$.

Vondrick et al. (2015) used a variation on classification images using invertible deep feature spaces. In order to avoid dataset bias introduced through perturbing real class exemplars, white noise in the feature space was used to generate stimuli. In this special case, category templates reduce to $n_A - n_B$. On each trial of the experiment, participants were asked to select which of two images (inverted from feature noise) most resembled a particular category. Because the feature vectors were random, thousands of images could be pre-inverted offline using methods that require access to large datasets. This early inversion method was applied to mean feature vectors for thousands of positive choices in the experiments and yielded qualitatively decipherable category template images, as well as better machine classification decision boundaries that were regularized by human bias. Under the assumption that human category distributions are Gaussian with equal variance, this method yields the vector that aligns with the line between means, although a massive number of pairs of random vectors (trials) are required.

Greene et al. (2014) conducted similar experiments in which a low-dimensional multi-scale gabor PCA basis was used to represent black-and-white images of scenes. Participants indicated which of two images was more similar to a seen image or a mental image of a scene. These judgments were used in an online genetic algorithm to lead participants to converge to their mental image. This method allowed for relatively efficient estimation of mental templates, although it is limited to scenes, lacks even the few theoretical guarantees of the Vondrick et al. (2015) method, and yields only a single meaningful image.

Finally, an alternative to classification images, Markov Chain Monte Carlo with People (MCMCP; Sanborn and Griffiths, 2007), constructs an experimental procedure by which humans act as a valid acceptance function in the Metropolis–Hastings algorithm, exploiting the fact that Luce's choice axiom, a well-known model of human choice behavior, is equivalent to the Barker acceptance function (see equation in Figure 1). On the first trial, a stimulus is drawn arbitrarily from the parameter space and compared to a new proposed stimulus that is nearby in that parameter space. The participant makes a forced choice of the better exemplar of some category (e.g., dog). If the initial stimulus is chosen, the Markov chain remains in that state. If the proposed stimulus is chosen, the chain moves to the proposed state. The process then repeats for as long as one wishes to run the sampling algorithm. MCMCP has been successfully employed to capture a number of different mental categories (Sanborn & Griffiths, 2007; Martin et al., 2012), and though these spaces are higher-dimensional than those in previous laboratory experiments, they are still relatively small and artificial compared to real images. Unlike classification images, this method makes no assumptions about the structure of the category distributions and thus can estimate means, variances, and higher order moments. Therefore we take it as a basic starting point for this paper.

### 2.2 DEEP NEURAL NETWORKS FOR IMAGES

Deep convolutional neural networks (CNN; LeCun et al., 1989) like AlexNet (Krizhevsky et al., 2012) are excellent for high-accuracy natural image classification and learn feature and category representations that are generally useful for a number of perceptual tasks. However, they can be difficult to interpret, especially in how they relate to human categorization behavior.

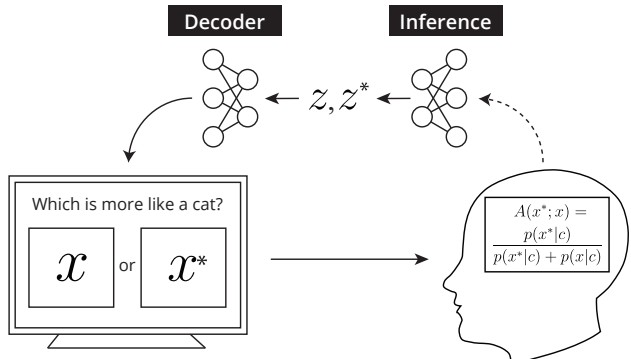

Figure 1: Deep MCMCP. A current state $z$ and proposal $z^*$ (top middle) are fed to a pretrained deep image generator/decoder network (top left). The corresponding decoded images $x$ and $x^*$ for the two states are presented to human raters on a computer screen (leftmost arrow and bottom left). Human raters then view the images in an experiment (bottom middle arrow) and act as part of an MCMC sampling loop, choosing between the two states/images in accordance with the Barker acceptance function (bottom right). The chosen image can then be sent to the inference network (rightmost arrow) and decoded in order to select the state for the next trial, however this step is unnecessary when we know exactly which states corresponds to which images. The inference network will, however, be needed later to characterize new images that we did not generate.

Generative Adversarial Networks (GANs; Goodfellow et al., 2014) and Variational Autoencoders (VAEs; Kingma & Welling, 2014) provide a generative approach to understanding image categories. In particular, these generative models provide an opportunity to examine how inference over a set of images and categories is performed. Generative models frame their approach theoretically on a Bayesian decomposition of image-label joint density, $p(z, x) = p(z)p(x|z)$, where $z$ and $x$ are random variables that represent a natural image distribution and its feature representation. Therfore $p(z)$ represents the distribution of natural images, and $p(x|z)$ is the distribution of feature vectors given an image. Together, $p(z)$ and $p(x|z)$ produce a high-dimensional constraint on image generation. This allows for easy visualization of the learned latent space, a property we exploit presently.

## 3 Markov Chain Monte Carlo in Deep Feature Spaces

In the typical MCMCP experiment, as explained above, the participant judges pairs of stimuli, and the chosen stimuli serve as samples from the category distribution of interest. This method is effective as long as noise can be added to dimensions in the stimulus parameter space to create meaningful changes in content. When viewing natural images for example, in the space of all pixel intensities, added noise is very unlikely to modify the image in interesting ways. We propose to instead perturb images in a deep feature space that captures only essential variation. We can then show participants images decoded from these feature representations in order to relate human judgments to the underlying latent space of interest, where category distributions are easier to learn. The resulting judgments (samples) approximate distributions that both derive arbitrary human category boundaries for natural images and can be sampled from to create new images, yielding new human-like generative image models. A schematic of this procedure is illustrated in Figure 1. Note that the method we propose does not require an inference network in practice, since we always know which latent code produced the images shown to the subjects. This is true as long as we initialize our MCMC chain states as random points in the latent space as opposed to in pixel space.

More specifically, popular deep generative networks such as GANs and VAEs learn a probability distribution $p(z)$ over some latent variable $z$, from which the objective of the network encourages of the mapping (generator or decoder) $M$ that $p(f(z; M)) = p(x)$, where $p(x)$ is the distribution over pixels in the dataset and $f$ is the family of deterministic functions s.t. $f : M \times z \to x$. If we assume that human psychological feature space $y$ results from some invertible transformation $T$ of $x$ (it is simply some learned transformation of raw input), then correspondingly $p(g(y; T^{-1}) = p(x)$, where $g$ is the family of functions s.t. $g : T^{-1} \times y \to x$. Because people in our experiments see only actual images $x$, perceived as $y$, the judgments they make are related to the deep latent space

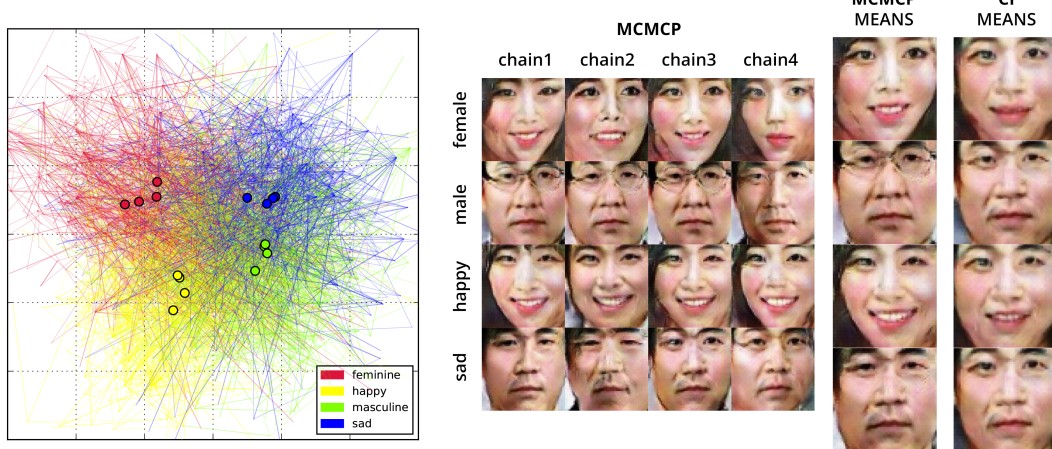

Figure 2: **Left:** Fisher Linear Discriminant projections of all four MCMCP chains for each of the four face categories. The four sets of chains overlap to some degree, but are also well-separated overall. Means of individual chains are closer to other means from the same class than to those of other classes. **Right:** Individual MCMCP chain means ($4 \times 4$ grid) and overall category means (second to last) visualized as images (overall CI means also shown for comparison in the final column). MCMCP means are much more differentiated than CI means, and better resemble the category in question.

through the following equation:

$$p(f(z; M)) = p(x) = p(g(y; T^{-1})). \tag{2}$$

Equation 2 assumes that humans approximate $p(x)$ analogously to an unsupervised learner, however, even if $T$ is a more specific feature representation supervised by classification pressure in the environment, we can assume information is only lost and not added, so that our deep generative models, if successful in their goal, will ultimately encode this information as a subset. Further, the equation above entails that $p(y)$ need not be identical to $p(z)$, and $M$ need not equal $T^{-1}$. However, for the practical sake of convergence using human participants, we would prefer that psychological space $y$ be similar to the deep latent space $z$ — we hope they contain the same relevant featural content for most images. While this assumption is hard to verify beyond face validity at this time, recent work suggests that some deep features spaces can be linearly transformed to approximate human psychological space (Peterson et al., 2016), and so we assume the deep latent space is relevant enough to human feature representations to act as a surrogate.

There are several theoretical advantages to our method over previous efforts. First, MCMCP can capture arbitrary distributions, so it is not as sensitive to the structure of the underlying low-dimensional feature space and should provide better category boundaries than classification images when required. This is important when using various deep features spaces that were learned with different constraints. MCMC inherently spends less time in low probability regions and should in theory waste fewer trials. Having generated the images online and as a function of the participant's decisions, there is no dataset or sampling bias, and auto-correlation can be addressed by removing temporally adjacent samples from the chain. Finally, using a deep generator provides drastically clearer samples than shallow reconstruction methods, and can be potentially be trained end-to-end with an inference network that allows us to categorize new images using the learned distribution.

## 4 HUMAN EXPERIMENTS

For our experiments, we explored two image generator networks trained on various datasets. Since even relatively low-dimensional deep image embeddings are large compared to controlled laboratory stimulus parameter spaces, we use a hybrid proposal distribution in which a Gaussian with a low variance is used with probability $P$ and a Gaussian with a high variance is used with probability

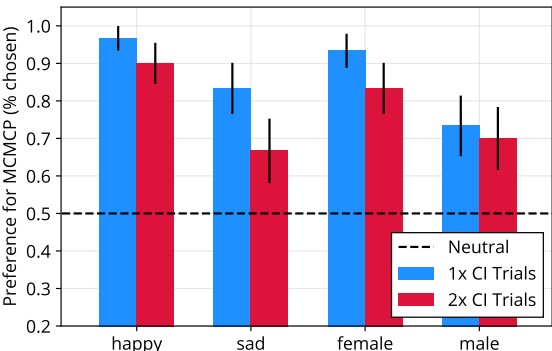

Figure 3: Human two-alternative forced-choice tasks reveal a strong preference for MCMCP means as representations of a category, when twice as many trials are used for CI.

$1 - P$. This allows participants to both refine and escape nearby modes, but is simple enough to avoid excessive experimental piloting that more advanced proposal methods often require.

Participants in all experiments completed exactly 64 trials (image comparisons), collectively taking about 5 minutes, containing segments of several chains for multiple categories. The order of the categories and chains within those categories were always interleaved. All experiments were conducted on Amazon Mechanical Turk. If a single image did not load for a single trial, the data for the subject undergoing that trial was completely discarded, and a new subject was recruited to continue on from the original chain state.

### 4.1 A First Proof of Concept: Face Categories

#### 4.1.1 Methods

We first test our method using DCGAN (Radford et al., 2015) trained on the Asian Faces Dataset. We chose this dataset because it requires a deep architecture to produce reasonable samples (unlike MNIST, for example), yet it is constrained enough to test-drive our method using a relatively simple latent space. Four chains for each of four categories (male, female, happy, and sad) were used. Proposals were generated from an isometric Gaussian with an SD of 0.25 50% of the time, and 2 otherwise. In addition, we conducted a baseline in which two new initial state proposals were drawn on every trial, and were independent of previous trials (classification images). The final dataset contained 50 participants and over $3,200$ trials (samples) in total for all chains. The baseline classification images (CI) dataset contained the same number of trials and participants.

#### 4.1.2 Results

MCMCP chains are visualized using Fisher Linear Discriminant Analysis in Figure 2, along with the resulting averages for each chain and each category. Chain means within a category show interesting variation, yet converge to very similar regions in the latent space as expected. Also on the right of Figure 2 are visualizations of the mean faces for both methods in the final two columns. MCMCP means appear to have converged quickly, whereas CI means only show a moderate resemblance to their corresponding category (e.g., the MCMCP mean for "happy" is fully smiling, while the CI mean only barely reveals teeth). All four CI means appear closer to a mean face, which is what one would expect from averages of noise. We validated this improvement with a human experiment in which 30 participants made forced choices between CI and MCMCP means. The results are reported in Figure 3. MCMCP means are consistently highly preferred as representations of each category as compared to CI. This remained true even when an additional 50 participants (total of 100) were run on the CI task, obtaining twice as many image comparison trials as MCMCP.

### 4.2 Larger Networks & Larger Spaces

The results in the previous section show that reasonable category templates can be obtained using our method, yet the complexity of the stimulus space used does not rival that of large object classification networks. In this section, we tackle a more challenging (and interesting) form of the problem. To

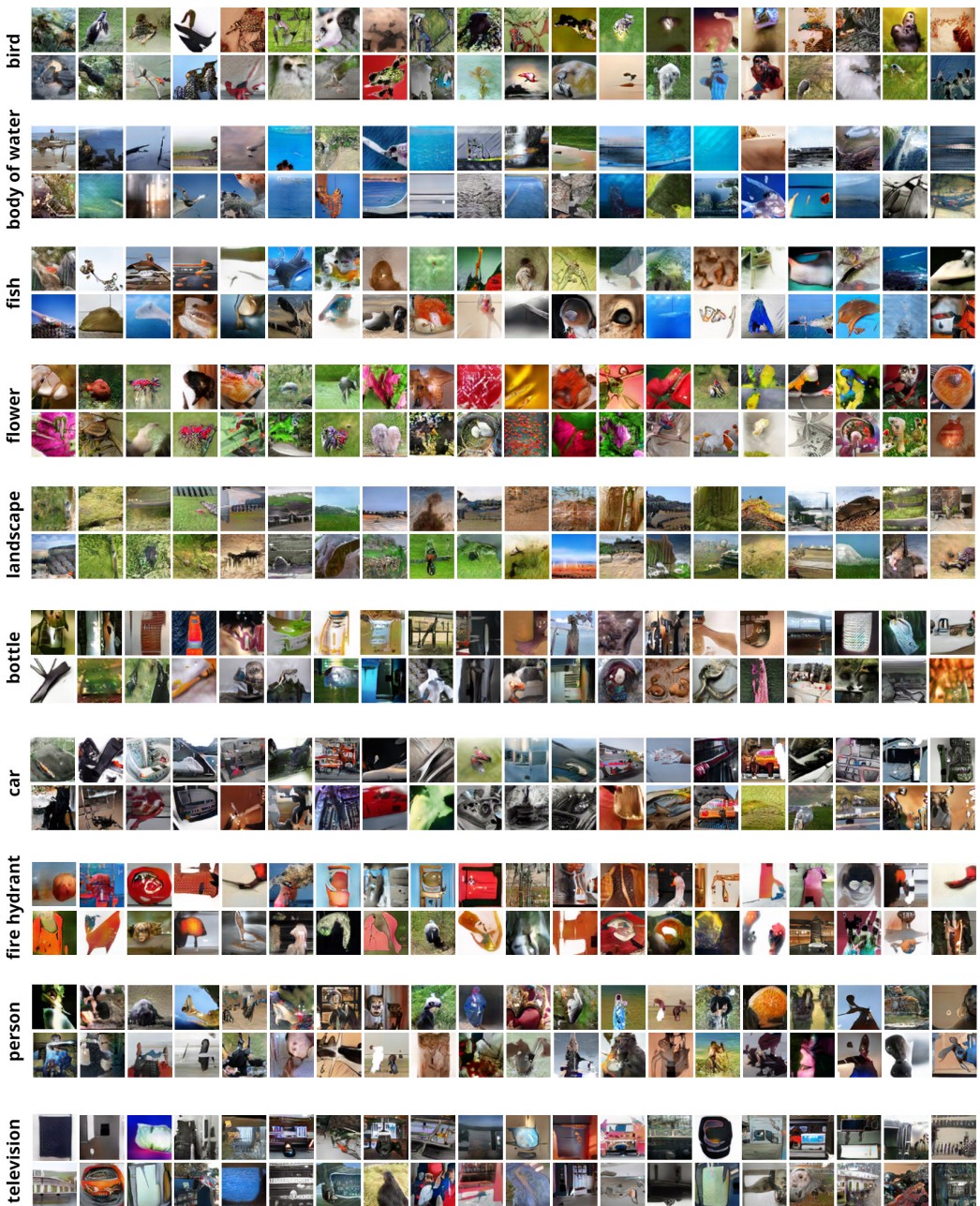

Figure 4: Most interpretable mixture component means (modes) taken from the top 40

do this, we employ a bidirectional generative adversarial network (BiGAN; Donahue et al., 2016) trained on the entire 1.2 million-image ILSVRC12 dataset ($64 \times 64$ center-cropped). BiGAN includes an inference network, which regularizes the rest of the model and produces unconditional samples competitive with the state-of-the-art. This also allows for the later possibility of comparing human distributions with other networks as well as assessing machine classification performance with new images based on the granular human biases captured.

### 4.2.1 METHODS

Our generator network was trained given uniform rather than Gaussian noise, which allows us to guarantee participants cannot get lost in extremely improbable regions. Additionally, we avoid proposing states outside of this hypercube by forcing $z$ to wrap around (proposals that travel out-

side of $z$ are injected back in from the opposite direction by the amount originally exceeded). In particular, we run our MCMC chains through an unbounded state space by redefining each bounded dimension $z_k$ as

$$z_k' = \begin{cases} -sgn(z_k) \times [1 - (z_k - \lfloor z_k \rfloor)], & \text{if } |z| > 1 \\ z_k, & \text{otherwise.} \end{cases} \tag{3}$$

Proposals were generated from an isometric Gaussian with an SD of $0.1$ 60% of the time, and $0.7$ otherwise.

We use this network to obtain large chains for two groups of five categories. Group1 included *bottle*, *car*, *fire hydrant*, and *person*, *television*, following Vondrick et al. (2015). Group2 included *bird*, *body of water*, *fish*, *flower*, and *landscape*. Each chain was approximately $1,040$ states long, and four of these chains were used for each category (approximately $4,160$). In total, across both groups of categories, we obtained exactly $41,600$ samples from $650$ participants.

To demonstrate the efficiency and flexibility of our method compared to alternatives, we obtained an equivalent number of trials for all categories using the variant of classification images introduced in Vondrick et al. (2015), with the exception that we used our BiGAN generator instead of the offline inversion previously used. This also serves as an important baseline against which to quantitatively evaluate our method because it estimates the simplest possible template.

### 4.2.2 RESULTS

The acceptance rate was approximately $50\%$ for both category groups, which is near the common goal for MCMCP experiments. The samples for all ten categories are shown in Figure 5B and D using Fisher Linear Discriminant Analysis. Similar to the face chains, the four chains for each category converge to similar regions in space, largely away from other categories. In contrast, classification images shows little separation with so few trials (5C and D). Previous work suggests that at least an order of magnitude higher number of comparisons may be needed for satisfactory estimation of category means. Our method estimates well-separated category means in a manageable number of trials, allowing for the method to scale greatly. This makes sense given that CI proposes comparisons between arbitrary images, potentially wasting many trials, and clearly suffers from a great deal of noise.

Beyond yielding our decision rule, our method additionally produces a density estimate of the entire category distribution. In classification images, only mean template images can be viewed, while we are able to visualize several modes in the category distribution. Figure 4 visualizes these modes using the means of each component in a mixture of Gaussians density estimate. This produces realistic-looking multi-modal mental category templates, which to our knowledge has never been accomplished with respect to natural image categories.

### 4.3 USING ESTIMATED HUMAN CATEGORIES FOR CLASSIFICATION

We also provide a quantitative assessment of the samples we obtained and compare them to classification images (CI) using an external classification task. To do this, we scraped approximately $500$ images from Flickr for each of our ten categories, which was used for a classification task. To classify the images using our human-derived samples, we used (1) the nearest-mean decision rule, and (2) a decision rule based on the highest log-probability given by our ten density estimates. For classification images, only a nearest-mean decision rule can be tested.

In all cases, decision rules based on our MCMCP-obtained samples overall outperform a nearest-mean decision rule using classification images (see Table 1). In category group 1, the MCMCP density performed best and was more even across classes. In category group 2, nearest-mean using our MCMCP samples did much better than a density estimate or CI-based nearest-mean.

## 5 DISCUSSION & CONCLUSION

Our results demonstrate the potential of our method, which leverages both psychological methods and deep surrogate representations to make the problem of capturing human category representations tractable. The flexibility of our method in fitting arbitrary generative models allows us to visualize

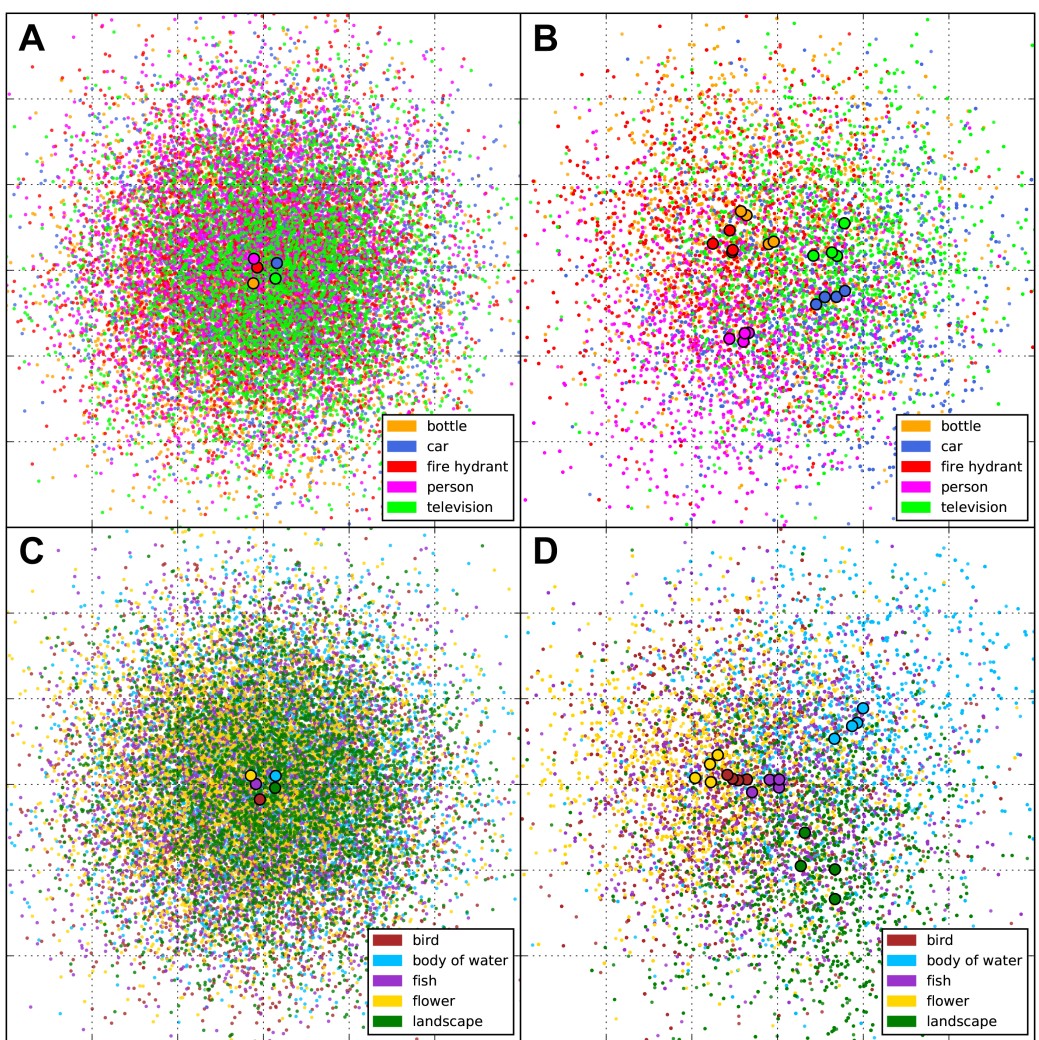

Figure 5: Fisher Linear Discriminant projections of **A**. Classification images comparisons for each category of group 1, **B**. samples for MCMCP chains for category group 1, **C**. Classification images comparisons for each category of group 2, and **D**. samples for MCMCP chains for category group 2. For A and C, large dots represent category means. For B and D, large dots represent chain means.

multi-modal category templates for the first time, and improve on human-based classification performance benchmarks. It is difficult to guarantee that our chains explored enough of the relevant space to actually capture the concepts in their entirety, but the diversity in the modes visualized and the improvement in class separation achieved are positive indications that we are on the right track. Further, the framework we present can be straightforwardly improved as generative image models advance, and a number of known methods for improving the speed, reach, and accuracy of MCMC algorithms can be applied to MCMCP make better use of costly human trials.

There are several obvious limitations of our method. First, the structure of the underlying feature spaces used may either lack the expressiveness (some features may be missing) or the constraints (too many irrelevant features or possible images wastes too many trials) needed to map all characteristics of human mental categories in a practical number of trials. Even well-behaved spaces are very large and require many trials to reach convergence. Addressing this will require continuing exploration of a variety of generative image models. We see our work are as part of an iterative refinement process that can yield more granular human observations and inform new deep network

Table 1: Classification performance compared to chance for both category sets (chance is 0.20).

|  | bird | body of water | fish | flower | landscape | all |
|---|---|---|---|---|---|---|
| MCMCP Mean | 0.33 | 0.28 | 0.01 | 0.57 | 0.67 | 0.37 |
| **MCMCP Density** | 0.23 | 0.31 | 0.18 | 0.44 | 0.73 | **0.38** |
| CI Mean | 0.23 | 0.30 | 0.2 | 0.24 | 0.52 | 0.30 |
|  | bottle | car | fire hydrant | person | television | all |
| **MCMCP Mean** | 0.15 | 0.32 | 0.11 | 0.77 | 0.73 | **0.42** |
| MCMCP Density | 0.25 | 0.56 | 0.26 | 0.19 | 0.50 | 0.35 |
| CI Mean | 0.28 | 0.62 | 0.15 | 0.12 | 0.13 | 0.26 |

CI = classification images

objectives and architectures, both of which may yet converge on a proper, yet tractable model of real-world human categorization.

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
