# OpenReview forum: "Capturing Human Category Representations by Sampling in Deep Feature Spaces"
_ICLR.cc/2018/Conference — Invite to Workshop Track_

### Official Review · AnonReviewer2 · 2017-11-24
**deep extension of MCMCP**

**Rating:** 6
**Confidence:** 4

**Review:**

Quality

This paper demonstrates that human category representations can be inferred by sampling deep feature spaces. The idea is an extension of the earlier developed MCMC with people approach where samples are drawn in the latent space of a DCGAN and a BiGAN. The approach is thoroughly validated using two online behavioural experiments.

Clarity

The rationale is clear and the results are straightforward to interpret. In Section 4.2.1 statements on resemblance and closeness to mean faces could be tested. Last sentences on page 7 are hard to parse. The final sentence probably relates back to the CI approach. A few typos.

Originality

The approach is a straightforward extension of the MCMCP approach using generative models.

Significance

The approach improves on previous category estimation approaches by embracing the expressiveness of recent generative models. Extensive experiments demonstrate the usefulness of the approach.

Pros

Useful extension of an important technique backed up by behavioural experiments.

Cons

Does not provide new theory but combines existing ideas in a new manner.

---

> ### Author Response · Authors · 2018-01-05
> **Small fixes and a note on theory**
>
> Thank you for your comments. We have addressed some typos and unclear sentences and agree that additional experiments in the future to understand the nature of the gendered smile bias face results would be interesting.
>
> Note that our work can be viewed as engaging with the theoretical problem of estimating unobservable mental content. MCMCP in pixel space provides the perfect solution to this problem, yet is surely intractable. Here we propose that a tractable first step is to assume a reasonable approximation (using an invertible feature space), from which further iterative improvements can be made.

---

### Official Review · AnonReviewer1 · 2017-11-27
**Using GANs for visualizing human representations of visual categories.**

**Rating:** 5
**Confidence:** 5

**Review:**

This paper presents a method based on GANs for visualizing how humans represent visual categories. Authors perform experiments on two datasets: Asian Faces Dataset and ImageNet Large Scale Recognition Challenge dataset.

Positive aspects:
+ The idea of using GANs for this goal is smart and interesting
+ The results seem interesting too

Weaknesses:
- Some aspects of the paper are not clear and presentation needs improvement.
- I miss a clearer results comparison with previous methods, like Vondrick et al. 2015.

Specific comments and questions:

-  Figure 1 is not clear. Authors should clarify how they use the inference network and what the two arrows from this inference network represent.
- Figure 2 is also not clear. Just the FLD projections of the MCMCP chains are difficult to interpret. The legend of the figure is too tiny. The right part of the figure should be better described in the text or in the caption, I don't understand well what this illustrates.
- Regarding to the human experiments with AMT: how do the authors deal with noise on the workers performance? Is any qualification task used? What are the instructions given to the workers?
- In section 4.2. the authors state "We also simultaneously learn a corresponding inference network, .... granular human biases captured". This seems interesting but I didn't find any result on that in the paper. Can you give more details or refer to where in the paper it is discussed/tested?
- Figure 4 shows "most interpretable mixture components".  How this "most interpretable" were selected?
- In second paragraph Section 4.3, it should be Table 1 instead of Figure 1.
- It would be interesting to see a discussion on why MCMCP Density is better for group 1 and MCMCP Mean is better for group 2. To see the confusion matrixes could be useful.

I like this paper. The addressed problem is challenging and the proposed idea seems interesting.  However, the aspects mentioned make me think the paper needs some improvements to be published.

---

> ### Author Response · Authors · 2018-01-05
> **All concerns have been addressed**
>
> Thank you for your comments and suggestions. We include many comparisons with the classification image method used by Vondrick et al. (2015) that focus on the mental content of the captured distributions, which is the goal of our paper (see Figures 2, 3, and 5, as well as Table 1). Like Vondrick et al. (2015), we show that classifiers derived from mental distributions do better than chance in predicting labels of real images. However, unlike Vondrick et al. (2015), note that we are not interested in augmenting computer vision methods to improve benchmark scores, but rather in developing innovative methods for modeling human mental representations.
>
> Answers to specific comments and questions:
>
> - Our newest draft makes Figure 1 more informative. Note that the inference network does not need to be used, and was not used in Section 4.1, because we know which z vectors generated which images for each set of trials, and do not need to convert the generated images back to inferred z representations in practice. However, the inference network is necessary for any application of our method that requires the use of any image not rendered by the network (i.e., in order to classify new images such as in Section 4.3).
>
> - Figure 2 has been enlarged and the newest uploaded draft extends the caption. The FLD projections simply show that the chains for different categories are well-separated, meaning that they successfully characterize different featural content.
>
> - We told AMT workers that it was important that they answer as best they can and used stringent selection criteria. If a single image did not load, the data was thrown out and a new subject was recruited to continue the chain at its original entry point.
>
> - In order to include inference in our GAN network, we use BiGAN (Donahue et al., 2016).
>
> - In Figure 4, we show the means of the mixture components with the largest mixture weights. We excluded only a small set of components that were presumably useful in explaining holdout samples and classifying, but which had no discernable visual content (washed out brown color). This appears to happen whenever large numbers of samples are summarized by a single mean (the CI method often showed only this behavior).
>
> - Comparing MCMCP Density and MCMCP Mean for groups 1 and 2 tells us little because the individual categories do not always give the same results. More importantly, we see the variation in results as a lesson that an inflexible method may not be able to cope with particular categories and how they interact with the particular latent space learned by the network. Using MCMCP avoids having to make any such limiting assumptions, and we can simply choose the density with the best fit to human samples.

---

### Official Review · AnonReviewer3 · 2017-11-27
**This paper proposes a method for using MCMCP to characterize a distribution of GAN-constrained image statistics corresponding to a human concept category. The paper describes a well-motivated method for investigating an interesting problem, but feels incomplete in exploring natural questions presented by the problem.**

**Rating:** 5
**Confidence:** 4

**Review:**

The idea of using MCMCP with GANs is well-motivated and well-presented
in the paper, and the approach is new as far as I know.  Figures 3 and 5 are
convincing evidence that MCMCP compares favorably to direct sampling of
the GAN feature space using the classification images approach.

However, as discussed in the introduction, the reason an efficient
sampling method might be interesting would be to provide insight
on the components of perception.  On these insights, the paper felt
incomplete.

For example, it was not investigated whether the method identifies
classification features that generalize.  The faces experiment is
similar to previous work done by Martin (2011) and Kontsevich
(2004) but unlike that previous work does not investgiate whether
classification features have been identified that can be added to an
arbitrary image to change the attribute "happy vs sad" or "male vs female".

Similarly, the second experiment in Table 1 compares classification
accuracy between different sampling methods, but it does not provide
any comparison as done in Vondrick (2015) to a classifier trained
in a conventional way (such as an SVM), so it is difficult to discern
whether the learned distributions are informative.

Finally, the effect of choosing GAN features vs a more "naive" feature
space is not explored in detail.  For example, the GAN is trained
on an image data set with many birds and cars but not many
fire hydrants.  Is the method giving us a picture of this data set?

---

> ### Author Response · Authors · 2018-01-05
> **Reasons for the noted omissions**
>
> Thank you for your comments and suggestions. We agree that more can be done to inspect the nature of the solution obtained by combining MCMCP with modern generative networks, but we see this as future application of the overall toolset we’ve designed and demonstrated in the current work. The suggested method of using classification features to change image attributes assumes a linear/additive feature space. Since we can learn any distribution with MCMCP, these simple methods do not apply to the general case.
>
> One of the methods we used to compare Classification Images (CI) and MCMCP was to assess how learned mental distributions could predict class labels for images held out from the training set. However, it is important to note that any held out set of images suffers from the same dataset bias that we sought to avoid (see introduction). For this reason, while a better estimate of a mental concept may perform better than other methods in predicting held out sets, there is no guarantee that it will converge to a model that performs equally or better than classifiers trained on those biased datasets. In keeping with the specific goals of our paper, we included no such analysis in our paper. However, the reviewer may find it useful to know that classifiers trained on a similar training set to the held out images were more successful in predicting class labels for those held out images (the test set). Inspecting the samples from the captured mental distributions gives us good reason to believe mental and synthetic concepts are different because many images favored by humans appear a great deal more abstract than what would be expected from current generative models (e.g., see water bottle examples).
>
> It is unclear how stratification of classes in the datasets used to train our networks detract significantly from the results presented in our paper (i.e., it is unlikely to interact with our finding regarding the improvement over CI). Also note that we strategically avoided classes that are most disproportionately represented in the ILSVRC12 dataset, such as “dog”, which makes up more than 10% of the dataset.

---

### Author Response · Authors · 2018-01-05
**Post-review changes**

Changes in response to initial reviews: clarifications, fixed typos, extended figure captions, and a small revision to Figure 1.

---

### Decision · Program_Chairs · 2018-01-29
**ICLR 2018 Conference Acceptance Decision**

**Decision:**

Invite to Workshop Track

**Comment:**

This paper introduces a GAN-based framework for inferring human category representations. The reviewers agree that the idea is interesting and well-motivated, and the results are promising. The technical contribution is not significant, but nevertheless the paper combines existing ideas in an interesting way. The reviewers would also like to see some more work towards the direction of investigation of the results and extraction of insights, without which the paper feels somehow incomplete.